# Comminution of Polymetallic Nodules with a High-Pressure Water Jet

**DOI:** 10.3390/ma15228228

**Published:** 2022-11-19

**Authors:** Przemysław J. Borkowski, Tomasz Abramowski, Monika Szada-Borzyszkowska, Wiesław Szada-Borzyszkowski

**Affiliations:** 1Faculty of Mechanical Engineering, Wroclaw University of Technology, 50-370 Wroclaw, Poland; 2Faculty of Navigation, Maritime University of Szczecin, 70-500 Szczecin, Poland; 3Faculty of Mechanical Engineering, Koszalin University of Technology, 75-620 Koszalin, Poland; 4Branch of the KUT in Szczecinek, Koszalin University of Technology, 78-400 Szczecinek, Poland

**Keywords:** polymetallic nodules, nodules comminution, hydro-jet mill, high-pressure water jet

## Abstract

This paper presents an original method for the comminution of polymetallic nodules in a hydro-jet mill of our own design, having the water jet pressure in the range of 70–200 MPa. The best comminution results are ensured by equipping the mill with a water jet having a diameter d_w_ = 0.7 mm and a comminution-homogenization tube having a diameter d_h_ = 2.4 mm, and by setting a distance s = 10 mm from the outlet of the homogenization tube to the comminuting disc. Particles of concretions comminuted under such conditions are characterized by a fairly regular isometric shape and favorable morphology represented by intense development of the specific surface.

## 1. Introduction

Recent years have seen an intensive growth of global production capacities, in particular those related to the manufacturing of modern electronic devices or new types of batteries for electric cars. This growth has led to an explosive demand for strategic raw materials. Such a demand cannot be met by resources extracted from conventional mines [1,2] and necessitates the search for new deposits of such minerals, which are also found in significant amounts at the bottom of the oceans [3,4], in the form of polymetallic nodules [5,6], cobalt-rich ferromanganese crusts [7,8], and polymetallic massive sulphides deposits [9,10]. Furthermore, according to some studies [2], the production of batteries for electric cars from metals extracted from conventional fossils generates a higher carbon footprint that is several times higher than the production from polymetallic nodules extracted from the ocean depths. 

To date, submarine explorations have enabled most of the all-ocean basins to be penetrated in detail. Of 19 isolated oceanic fields [11], one of the richest and best-studied areas is the Clarion-Clipperton zone [12,13] lying in the northeastern equatorial Pacific. The number of polymetallic concretions deposited in this zone alone is estimated at 21 billion metric tons [14]. These concretions contain significantly more Mn, Te, Ni, Co, and Y, and in the case of Tl, this content is as much as 6000 times greater than the entire global base of known terrestrial resources of these metals. In comparison to global terrestrial resources, they also contain significant amounts of Cu, Mo, W, Li, Nb, and rare earth oxides [15]. In fact, the number of polymetallic nodules may be much higher than currently estimated due to the fact that their amount is determined by their content in the upper layers of the deposit [16], and its thickness is unknown.

Nowadays, cobalt-rich ferromanganese crusts [17], which also contain nickel, molybdenum, and copper [18], are also becoming increasingly important for the development of electronics and battery industries. In addition, such crusts sometimes also contain rare earth elements [19,20]. For example, Fe-Mn crusts found in the Clarion-Clipperton zone contain significantly more metals (Tl—1700 times more, Te—10 times more, as well as Co and Y) than the entire global terrestrial resource base of these metals. In contrast, the contents of other metals of interest (Bi, REO, Nb, and W), including the rare earth elements found in these crusts, are comparable to global terrestrial mineral reserves [6,12].

However, the extraction of such subsea deposits is complicated, as it requires innovative mining operations which, on the one hand, take into consideration various types of hazards, and on the other hand give priority to protecting sensitive environments of subsea ecosystems [21]. For example, the uncontrolled discharge of fine-particle wastes generated during the extraction and transport of nodules [13] spreads relatively easily in the form of ‘plumes’ of sediment-laden water [22,23]. This is particularly hazardous in the case of deep-sea toxic metal mining [24], as the released particles can cause considerable disturbance to marine environments across large areas.

Various methods have been used to effectively collect polymetallic nodules from the ocean floor: magnetic [25], mechanical [5,26], and hydraulic, which uses a high-pressure water jet [25,27]. Based on the results of their investigations [5,28], suitable mechanical [25,27], hydraulic [27,29], and hybrid [30,31] collectors have been designed, most commonly working in combination with a hydraulic transportation system consisting of a bottom flexible pipeline and a vertical rigid pipe [32,33].

Vertical transport of polymetallic nodules from the seabed is usually carried out using suitable pipelines [31,33] and following either a hydraulic [34,35], a hydro-pneumatic [33], or—less commonly—a pneumatic [36] method. However, due to the considerable depth of the nodule excavation site, such means of vertical transport are subject to numerous complications [37,38], especially in the flexible part of the pipeline [39,40]. For the above reasons, the search for a more efficient and reliable method of transporting ore [41] from underwater deep-sea mining sites is continued.

We believe that most of the negative phenomena interfering with the processes of extracting polymetallic nodules and their efficient pipeline transport are avoidable by appropriate fragmentation of the nodules at the site of extraction [42]. For example, in the case of extracting cobalt-rich ferromanganese crusts, such fragmentation cannot be avoided anyway [43]. Therefore, the aim of this study is to present the effects of comminution of polymetallic concretions using a high-pressure water jet considered in the context of facilitating transport to the surface as well as further processing.

## 2. Equipment, Apparatus, and Test Conditions

The study involved polymetallic nodules mined from depths of approximately 4200–4300 m below sea level in the Clarion-Clipperton part of the International InterOceanMetal (IOM) zone. The area is located in the northeastern equatorial Pacific about 1600 km west of Mexico.

The concretions were crushed in two passes, initially using a jaw crusher to obtain feed particles with the grain size of less than 2 mm. They were subjected to final comminution in a hydro-jet mill [44] of our own design, constructed as a result of analyzing many similar designs included in the most relevant US patents [45,46] as well as in publications [47,48] from China and other countries. The hydro-jet mill uses the energy of a high-pressure water jet by following an operating principle similar to that of an abrasive water jet in a cutting head [49,50]. 

A schematic diagram of the construction of such a mill is shown in Figure 1. Its grinding elements comprise a comminuting-homogenizing tube (4) and a comminuting disk (5), which are made of sintered tungsten carbide. The opening of the comminuting-homogenizing tube has an inlet part with a threshold edge, where the feed is initially comminuted. The final comminution of the ore is the result of collisions with the surface of the comminuting disk, which is static but can be rotated by an appropriate angle as its surface erodes. 

Basic efficiency tests of the final waterjet comminution of polymetallic concretions were carried out under the following conditions:-nominal water jet pressure p = 70, 100, 150, and 200 MPa,-water nozzle diameter d_w_ = 0.5, 0.6, 0.7, 0.8, and 0.9 mm,-diameter of the comminuting-homogenizing tube d_h_ = 1.6, 2.4, and 3.4 mm.

The effects of polymetallic nodule comminution were measured using an Analysette 22 MicroTec Plus laser particle size meter, enabling the analysis and production of particle size distributions in the range 0.08–2000 μm.

An FEI Quanta 200 Mark II scanning microscope (ThermoFisher Scientific Inc., MA, USA) was used to evaluate the particle shape and morphology of fragmented polymetallic nodules. In turn, the chemical composition of these nodules was identified with the EDAX Genesis XM 2i (AMETEK Inc., PA, USA) chemical analyzer, which is a specialized equipment extension of this microscope.

## 3. Crystallographic and Mineralogical Characteristics of Polymetallic Nodules

The general external view and the size of typical polymetallic nodules analyzed in this study are shown in Figure 2.

Most are rough nodules with ellipsoidal shapes similar to slightly flattened spheres. Preliminary studies show their relatively low resistance to mechanical chipping. A typical view of the chipping of such concretions and their cross-section resulting from the impact of a high-pressure abrasive water jet (AWJ) jet is shown in the representative Figure 3. Even from such cross-sections, it is evident that the layered structure of these concretions is a result of the conglomeration of large particles.

Such a multilayered internal structure of polymetallic nodules is clearly visible in the crystallographic scans, shown by means of example in Figure 4.

In the course of making the above preparations, the polymetallic nodules were found to be highly susceptible to disintegration and to have low strength. Therefore, even such preliminary studies confirm previous literature reports on the low mechanical strength [29,51] of nodules, which manifests itself in their impact fragmentation not only in centrifugal pumps [52] but also even due to mutual contacts as they are transported in hydraulic pipelines [34] to the surface, as well as during the testing of the samples [53]. All of the above demonstrate the high susceptibility of polymetallic concretions to dynamic crushing, especially under the influence of high-pressure water jets.

Studies of mineralogical composition were conducted on polymetallic concretions mined during the IOM-2014 cruise from a plot managed by the international organization InterOceanMetal, located in the Clarion-Clipperton zone in the Pacific Ocean. The concretions were mined using a dragnet on the H22 exploration block in the IOM exploration area from a depth of approximately 4200–4300 m below sea level. The studied polymetallic concretions were mostly ellipsoidal in shape, with the longest dimension ranging from 30 to 60 mm.

The mineralogical composition of the studied polymetallic concretions mainly contained oxides of various elements, mostly metallic. After separating the most common metallic elements, their average content in the polymetallic concretions was, respectively: 32.1% Mn, 6.7% Si, 5.7% Fe, 2.5% Al, 2.3% Na, 2% Mg, 1.7% Ca, 1.3% Cu, 1.3% Ni, 1.1% K, 0.2% Co. A graphical illustration of the chemical composition of the investigated polymetallic concretions is shown in Figure 5. Among the metals contained in these concretions, only cobalt was found in markedly smaller amounts.

Very similar results were also obtained in a much more extensive study of such nodules conducted at a specialized chemical analysis laboratory [54]. Moreover, the studies showed that the concretions contained rare earth metals in amounts that were insufficient to make their mining profitable in present conditions.

## 4. Preliminary Mechanical Grinding of Polymetallic Concretions

In order to facilitate the water-jetting comminution of polymetallic concretions, a procedure for their mechanical pre-shredding was required. In the present study, a jaw crusher, operating at a capacity of 30.56 kg/h, was used to pre-crush the concretions. Example images of particles obtained from the mechanical grinding of polymetallic concretions using such a crusher are shown in Figure 6.

A general view of polymetallic nodule particles crushed using a jaw crusher is shown in Figure 6a. The particles of crushed concretions are nodules of various sizes with irregular shapes, as they are crushed along the boundaries of the constituent particles, thus breaking numerous bridges of inter-particle conglomerates. Therefore, they are characterized by numerous areas of distinctly rough surface, characteristic of brittle decohesion, a typical example of which is the SEM image visualized in Figure 6b. In its lower and right part, extensive cracks are also visible, which are usually the beginnings of the boundaries of new, smaller particles to be formed in the particle during its further breakdown. Such susceptibility of these conglomerates, which usually comprise polymetallic concretions, results in the formation of a very large number of relatively fine particles during their grinding.

Such a method of preliminary crushing of polymetallic concretions provides approximately 23.8% by weight of particles with dimensions exceeding 1 mm, approximately 18.1% by weight of particles with dimensions of 0.7–1 mm, approximately 14.4% by weight of particles with dimensions of 0.43–0.7 mm, and approximately 43.7% by weight of the finest particles, with dimensions below 0.43 mm. The percentage composition of the size of particles formed from ground polymetallic concretions is shown in Figure 7. All such particles are the feed for their final comminution performed in a hydro-jet mill [44] under the action of a high-pressure water jet.

## 5. Determination of Optimum Parameters for Comminution of Polymetallic Nodules by Water Jet

The appropriate diameter of the water jet was determined by performing tests on the comminution efficiency of polymetallic nodules with various water jets. Illustrative results of such tests are shown in Figure 8 as a function of water pressure. 

The graphs show that the comminution efficiency of polymetallic nodules increases with the increasing water pressure and, above all, with the increasing diameter of the water nozzle. The comminution efficiency Q_n_ increases particularly with increased diameters of the smallest nozzles (d_w_ = 0.5 and 0.6 mm), which use relatively smaller amounts of water. The most favorable increases in the comminution efficiency of polymetallic nodules are obtained with water nozzles having a diameter of d_w_ = 0.7 mm. Therefore, a water nozzle with this optimum diameter was used in all further studies. 

On the other hand, the use of nozzles with larger diameters (d_w_ = 0.8 mm) results in an unfavorable excessive amount of water. In some cases, this excess water causes (for a nozzle with d_w_ = 0.9 mm) the mill to choke and lose efficiency.

In the subsequent research phase, the comminution efficiency of polymetallic nodules was tested in order to optimize the diameter of the comminuting-homogenizing tube. Examples of the test results are shown in Figure 9.

The analysis of these graphs shows that the comminution efficiency of the polymetallic nodules increases with increasing water jet pressure. At the same time, the intensity of this effect depends on the diameter of the comminuting-homogenizing tube. The tests demonstrate that in the area of the analyzed values, the most favorable performance is obtained with a comminuting-homogenizing tube having a diameter of d_h_ = 2.4 mm. For these reasons, a comminuting-homogenizing tube with this diameter was used in all other tests.

In the final phase of the research, attempts were made to assess the effect of the distance from the outlet of the comminuting-homogenizing tube to the comminuting disc. The average dimension a_90_ [µm] of the particles obtained in the comminution of polymetallic nodules, expressed as a function of the length s [mm] of the water jet, is illustrated in the graphs presented in Figure 10.

In the range of the smallest distances (s = 5 mm), a significant number of particles do not collide directly with the hard surface of the comminuting disk and, therefore the average values of the dimension a_90_ [µm] are relatively, high despite the intense turbulence occurring in the comminuting chamber of this mill. The most effective comminution of concretion particles occurs when a distance of s = 10 mm is ensured in all ranges of the applied water pressures. Such a distance has already been optimized for identical comminution of brittle materials of similar dynamic strength [44]. Increasing the distance further leads to unnecessary dissipation of the water jet energy and to a gradual increase in the size of the comminuted particles.

## 6. Comminution of Polymetallic Concretions with a Medium-High-Pressure Water Jet

The present attempt to use a high-pressure water jet for micro comminution is motivated by the search for an effective method of comminution of polymetallic concretions while ensuring the greatest possible development of the specific surface area of the obtained particles. Such parameters are required to intensify the radically different metallurgical processes [55,56] employed in the treatment of concretions, and are based on various methods [57,58] of bio-leaching and bioprocessing concretions with the participation of appropriate fungi [59].

Due to the low strength of polymetallic concretions, relatively low water jet pressures were used in the present study of their comminution. This decision was also guided by ecological considerations in order to limit the excessive comminution of concretions. According to ecological recommendations [60,61], nodule particles finer than ˂63 µm are considered hazardous waste [13,34], as they can cause extensive pollution of marine environment if they are ejected in the form of “plumes” of water containing such sediment [22,23].

The effects of such comminution carried out at moderately high-water jet pressures are illustrated by the example particle distributions shown in Figure 11.

When using a water jet with a pressure of 70 MPa (Figure 11a), as much as 90% of the particles of comminuted nodules are in the size range of 0.1–333 µm, and the remaining 10% of the mass of comminuted particles is in the range of 333–615 µm. It should be noted here that under such conditions, about 28% of the crushed particles are finer than ˂63µm, i.e., they are environmentally hazardous waste [13,62].

On the other hand, in the case when concretions are comminuted at a water pressure of 100 MPa (Figure 11b) the dimensions of the obtained particles are in the range of 0.04–211 µm and 10% of the largest particles do not exceed the dimension of 440 µm. Under such comminution conditions, the content of fine particles discussed above increases to the level of 40%. For the above reasons, the use of appropriate seals in technological equipment is essential to eliminate the impact of this waste on marine environments.

Analysis of the quality of the particles of polymetallic nodules comminuted with the high-pressure water jet clearly shows the rather common occurrence of isometrically shaped particles (Figure 12). In addition, it is noticeable that usually the dominant action of the high-pressure water jet results in the spatial development of the surface layers of polymetallic nodules on the surface of the crushed concretion particles. Occasionally, it is also possible to observe (e.g., in Figure 12b) deformation of the surface of the concretion particles produced by mechanical impacts, characterized by distinct metallic reflections, which, in fact, is a visual confirmation of the high metal content.

Analogous phenomena also occur during the comminution of polymetallic concretions at a higher water jet pressure (Figure 13). It should be noted here that with increasing water pressure, the concretions disintegrate into finer particles and their surface layers assume increasingly developed spatial forms.

As increasing water jet pressure was observed to have a relatively positive effect on the comminution intensity of polymetallic concretions and on the shape of the resulting particles, we decided to also conduct analogous studies at higher water pressures.

## 7. Comminution of Polymetallic Concretions with a Water Jet at Very High Pressures

The effects of comminuting polymetallic concretions in a hydro-jet mill using water jets at very high pressures are illustrated by the example particle distributions shown in Figure 14.

The use of a water jet with a pressure of 150 MPa results in 90% of the particles of crushed concretions having dimensions within the range of 0.04–129 µm (see Figure 14a) and the remaining 10% of larger particles not exceeding the dimension of 252 µm. Under such hydrodynamic conditions, about 64% of comminuted nodule particles reach dimensions smaller than ˂63 µm, which, for marine environmental reasons, is considered the hazardous limit size value [13] for such waste.

Even finer-grained particles are obtained when concretions are comminuted with a water jet at 200 MPa (Figure 14b). The dimensions of 90% of the obtained particles are in the range of 0.04–75 µm and 10% of the remaining particles are in the range of 75–185 µm. Under such conditions, the content of dangerously fine concretion particles (˂63 µm) increases to a value of 87%. In addition, the application of the highest water pressures results in a bi-modal shape of the size distribution of the crushed particles (Figure 14b). This fact is related to the specific properties of the finest concretion particles, which are the subject of a separate study.

In addition, the application of a water jet with increasingly higher pressures was found to cause more intensive comminution of polymetallic concretions particles. This is clearly evident when comparing the SEM images shown in Figure 15a and Figure 16a.

At lower image magnification values, the individual particles seem to have quite regular isometric shapes, which is true when referring to their macro scale. Fundamental differences occur only at high magnification values, on the order of 5000 times. In such case it can be observed (Figure 15b) that the water jet with the pressure of 150 MPa dynamically washed out the less resistant particles of the concretions, forming highly developed surfaces on their surface layer.

This is particularly visible in SEM images illustrating the multilevel development of the particle surface layers produced when the concretions are comminuted with the 200 MPa water jet (Figure 16b). Under such conditions, the relative unit area of such particles is much more developed compared to the area of particles crushed at lower water pressures.

## 8. Results Discussion

The analysis of the results of the above study indicates that an increase in the water pressure significantly increases the power of a high-pressure water jet. The average values of the power of such a water jet which occur in the studied range of pressures used in a mill of this type are described by the following empirical relationship:N = 0.0005p^2^ + 0.01219p − 2.4039  (for R^2^ = 1)(1)

Importantly, the comminution process of polymetallic nodules is characterized not only by the power of the water jet, but also by the specific energy input required in this process. This specific energy consumption of nodule comminution in a waterjet mill can be determined from the following relationship:(2)$$\;E≅\frac{p\cdot Q_{w}\cdot \;t}{m_{n}}$$

Although the water jet in such a mill has a relatively high efficiency (on the order of 83%), some of its energy is lost, resulting in an increase of the unit energy input consumed in the comminution process of polymetallic concretions. Therefore, in the scope of the present research, the average values of the unit expenditures of comminution energy as a function of water pressure are described by the following relationship:E = 0.00002p^2^ + 0.0061p + 0.6269  (for R^2^ = 0.9993)(3)

Both relationships are best illustrated by the graphs in Figure 17.

In order to determine the energy consumption of the hydro-jet comminution process of polymetallic nodules as a function of the water jet diameter, appropriate tests were carried out, and their illustrative results are presented, in relationship to the applied water jet pressures, in Figure 18.

All of the curves representing the test results have a similar shape. On each occasion, the smallest values of specific energy inputs are observed for water jets with a diameter of d_w_ = 0.7 mm. At lower water pressures, the curve has a more flattened shape, whereas the slopes become steeper as the water pressure increases. All of the graphs clearly confirm that the above water nozzle diameter is optimal.

The following scatter of the results of individual parallel measurements, shown by means of example in Figure 19, may also serve as an additional partial illustration to the issues of the energy intensity of the comminution process of polymetallic nodules in such a hydro-jet mill.

The histogram clearly shows that an increase in the water jet pressure results in an increase both of the unit energy expenditure of the waterjet comminution process and of the spread of the values of individual parallel measurements. This is evidenced by the increasing heights of the confidence intervals, represented as red arrows.

In evaluating the comminution process of polymetallic concretions, it is also important to know the relative spreads of unit energy inputs occurring when different water jet pressures are applied. The relative values of such spreads occurring in the confidence intervals of the individual test series were determined according to the following formula:(4)$$\text{±}\Delta E=\left[\frac{\left(E_{\max}-E_{\min}\right)\;}{2\;E}\right]\;100\;[\%]$$

The relative spread values of such unit energy inputs, determined according to relationship (3), are illustrated by the histogram in Figure 20.

The graph clearly shows that an increase in water pressure produces a considerable decrease in the spread of the relative values of such unit energy inputs. Thus, in such a hydro-jet mill, an increase in water pressure has a beneficial effect on stabilizing the comminution process of polymetallic concretions.

Such a water jet method of concretion comminution generally offers much higher efficiency than mechanical comminution does. However, the specific energy values in the case of waterjet comminution significantly exceed the analogous energy values needed in mechanical grinding.

The results presented in the previous chapters clearly indicate that increasing the pressure of the water jet intensifies the comminution of polymetallic nodules, causing a decrease in the size of the obtained particles. In order to facilitate visual comparison of the particle size of concretions formed during the comminution with water jets at different pressures, the corresponding SEM images with equal magnification (500×) are summarized in Figure 21.

The content of particles smaller than 63µm, being the hazardous limit size value [13] of such wastes due to marine environmental protection [63,64], also changes to an analogous degree. At a water jet with a pressure of 70 MPa, their content among crushed nodules is about 28%, and at 100 MPa it is 40%. When comminuted with a water jet at a pressure of 150 MPa, their content increases to about 64% of the particles, and at a pressure of 200 MPa it is as high as 87%. Thus, with a certain simplification, it can be concluded that the comminution of polymetallic concretions with a water jet at very high pressures causes excessive fragmentation and primarily produces waste which is harmful to the marine environment.

Under such conditions, and especially at the highest water pressures and in the absence of effective sealing of the deep-sea processing equipment, the ejaculation of “plumes” of sediment-laden water can cause extensive environmental pollution [22]. Nevertheless, even the finest (˂20 µm) sediments are not always dangerous. If such particles have a developed surface, under typical marine conditions they tend to fluctuate rapidly. This usually results in the formation of large aggregates with diameters reaching up to 1100 μm [23].

Thus, if the dimensions criterion for the undersized particles was corrected to a limit of ˂20 µm, then the products from our hydro-jet mill at water pressures of 70, 100, 150, and 200 MPa would contain, respectively, 10, 13, 25, and 46% of excessively comminuted particles.

Moreover, as evident from complementary analyses of issues related to deep sea mining (DSM), in mining practices some technological enrichment operations will be carried out directly at sea [42,43].

The results presented in Chapter 6 show that in a hydro-jet mill, the maximum dimensions for 90% of nodule particles comminuted at the water pressure p = 70 MPa, are a_90_ = 333 µm, and for p = 100 MPa, a_90_ = 211 µm, respectively. On the other hand, according to the analogous data in Chapter 6, the dimensions of the particles of concretions comminuted at the pressure p = 150 MPa are: a_90_ = 129 µm, and for p = 200 MPa, a_90_ = 75 µm. Thus, in such a study range of the comminution of polymetallic concretions, the average dependence of the dimension a_90_ of the obtained particles as a function of the pressure of the water jet is described by the following relationship:a_90_ = 0.014 p^2^ − 5.6906 p + 655.12  (for R^2^ = 0.9883)(5)

In turn, an increase in the water jet pressure results in an increase in the efficiency of the comminution process of polymetallic concretions. This is due to the increase in both water output and the velocity of the particles of the comminuted concretions, which induces an increase in their kinetic energy. Therefore, at the water pressure of 70 MPa, the efficiency of the comminution of concretions in the hydro-jet mill is 26.57 kg/h and at 100 MPa the comminution efficiency is 37.69 kg/h. Increasing the water pressure to 150 MPa increases the comminution efficiency to 49.03 kg/h, while for p = 200 MPa the comminution efficiency reaches 58.00 kg/h. Therefore, in such a study range of the comminution of polymetallic concretions, the average value of the efficiency of the water jet comminution process is described by the following relationship:Q_n_ = −0.0009 p^2^ + 0.4872 p − 2.6013  (for R^2^ = 0.9977)(6)

The above relationships are best illustrated by the graphs in Figure 22.

Figure 23 shows the differences between the efficiencies of the comminution of polymetallic concretions using water jets of different pressures. These data confirm previous observations that an increase in water pressure results in increased efficiency of the waterjet comminution process, and moderately contributes to an increase in the spread of values of individual parallel measurements. This is evidenced by the increasing heights of the confidence intervals, marked as red arrows.

On the other hand, somewhat different relationships are found when considering the relative spread of the efficiencies of the comminution process, occurring at different water jet pressures. The relative spread values these efficiencies occurring within the confidence intervals of the individual test series were determined according to the following formula:(7)$$\text{±}\Delta Q_{n}=\left[\frac{\left(Q_{\max}-Q_{\min}\right)\;}{2{\;Q}_{n}}\right]\;100\;[\%]$$

The relative spread values of the comminution efficiency of polymetallic concretions, determined according to relationship (6), are illustrated by the histogram in Figure 24.

The analysis of the above graph demonstrates that with an increase in the water pressure, the relative spread values of the comminution efficiency of concretions are significantly reduced. Thus, an increase in water pressure facilitates the comminution process of concretions in a hydro-jet mill.

Taking into account all the physical and technical parameters of the high-pressure water jet and the criteria characterizing its suitability for the final comminution of polymetallic concretions in a hydro-jet mill with the following working parameters: d_w_ = 0.7 mm, d_h_ = 2.4 mm, s = 10 mm, we concluded that the most favorable comminution effects are obtained with the water jet pressure of 100 MPa. In this case, the power of such a water jet reaches the value N = 14.48 kW and the efficiency of the comminution process of polymetallic concretions is Q_n_ = 37.69 kg/h, with unit energy input required for the comminution of polymetallic concretions being E = 1.39 MJ/kg. This configuration results in particles of comminuted polymetallic nodules having a_90_ = 211 µm and an isometric shape with a relatively significantly developed specific surface area. Of these, approximately 40% are excessively fine particles with dimensions smaller than ˂63 µm, which is considered to be the hazardous limit [13] size value for reasons of marine environmental protection [62,64]. As, presently, this limit is also being shifted to ˂20 µm [23], the nodule particles comminuted in our hydro-jet mill contain only 13% of excessively fine particles.

Such a concept of sub-sea comminution of polymetallic nodules extremely simplifies the issue of hydro-pneumatic transportation of segregated ore to the surface. However, it imposes the necessity to prepare a suitable storage site for technological waste on the seabed [65,66] and to protect it from being washed away.

If the development of DSM technology is carried out according to the currently accepted principles, i.e., extracting polymetallic nodules to the surface and subjecting them to safe processing, the problem of managing the waste generated in these processes may even disappear, as these multidirectional ecological issues have long been studied and fully developed [67].

## 9. Conclusions

Detailed analyses of the results of the present study enabled the formulation of the following conclusions:-Effective comminution of polymetallic concretions was possible in a hydro-jet mill of our own design, in which a water jet was applied with the pressure varying in the range of 70–200 MPa. The best comminution results were ensured by using the following mill parameters: water jet having a diameter d_w_ = 0.7 mm, comminution-homogenization tube having a diameter d_h_ = 2.4 mm, distance of the homogenization tube outlet from the comminution disc s = 10 mm.-The comminution of polymetallic concretions in the hydro-jet mill generally produces particles characterized by a fairly regular isometric shape and a favorable morphology in the form of an intensively developed specific surface.-Increasing the pressure of the water jet, which causes an increase in its power, also improves the effectiveness of the increasingly intensive comminution of polymetallic concretions. It also increases the specific surface area of the produced particles, and thus the efficiency of such a comminution process. However, in terms of the energy intensity of the comminution process, the increase in water pressure is unfavorable, although it should be noted that the spread of the relative values of the unit energy inputs is significantly reduced, being beneficial for the stabilization of the process.-An increase in the water jet pressure {p [MPa]} of the hydro-jet mill with optimal parameters results in an increase in its power output according to the following empirical relationship: N = 0.0005p^2^ + 0.01219 p − 2.4039 [kW].-As the pressure of the water jet {p [MPa]} used in the hydro-jet mill with optimal parameters increases, the specific energy input necessary for the comminution of polymetallic concretions increases according to the following empirical relationship: E = 0.00002p^2^ + 0.0061p + 0.6269 [MJ/kg].-The efficiency of the water jet comminution of polymetallic concretions, determined by the average size of 90% of the comminuted particle content {a_90_ [µm]}, is described as a function of water pressure {p [MPa]} by the following empirical relationship: a_90_ = 0.014p^2^ − 5.6906p + 655.12 [µm].-An increase in the water jet pressure {p [MPa]} of the hydro-jet mill with optimal parameters increases the efficiency of the comminution process of polymetallic concretions according to the following empirical relationship: Q_n_ = −0.0009p^2^ + 0.4872p − 2.6013 [kg/h].-With all of the above criteria taken into consideration, the most favorable effects of the comminution of polymetallic concretions in a hydro-jet mill were found to be obtained at the water pressure of 100 MPa. At such pressure, the power of the water jet reaches the value of N = 14.48 kW and the efficiency of the comminution process of polymetallic nodules is Q_n_ = 37.69 kg/h, with the unit energy input of E = 1.39 MJ/kg. This configuration results in particles of crushed polymetallic nodules having a_90_ = 211 µm and an isometric shape with a relatively well-developed specific surface area. Of these, 40% are undersized particles, with dimensions ˂63 µm. However, with a more useful criterion used and the limit corrected to ˂20 µm, the content of undersized particles is only 13%.-For the above reasons, the water jet method of comminution of such concretions is very promising.-The current design of the universal hydro-jet mill used in this research provides a developmental basis for improving DSM technology.

## Figures and Tables

**Figure 1 materials-15-08228-f001:**
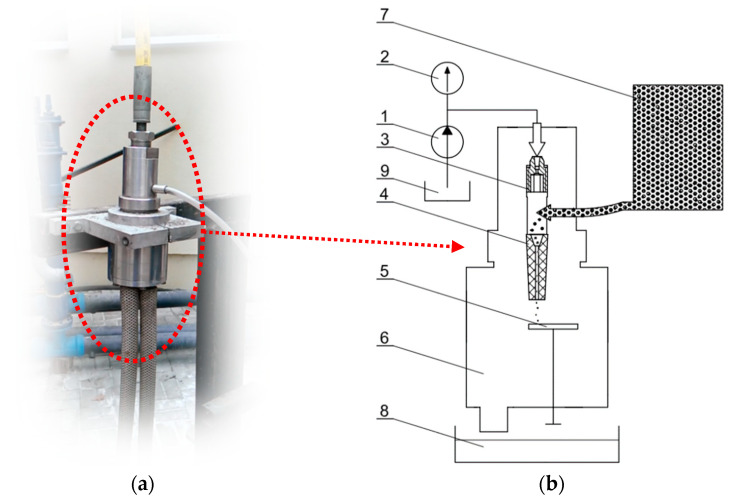
General view of experimental setup (**a**), and diagram of a hydro-jet mill (**b**): 1—high-pressure pump, 2—manometer, 3—water tube, 4—comminuting-homogenizing tube, 5—comminuting disk, 6—comminuting chamber, 7—feed tank, 8—product tank, 9—water tank.

**Figure 2 materials-15-08228-f002:**
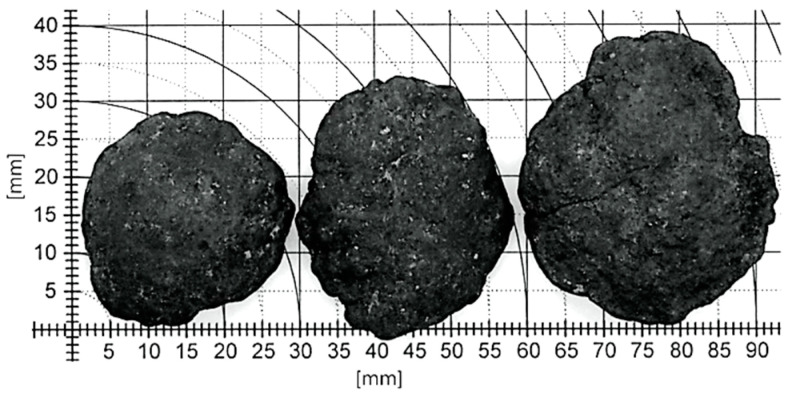
Examples of external views of polymetallic nodules.

**Figure 3 materials-15-08228-f003:**
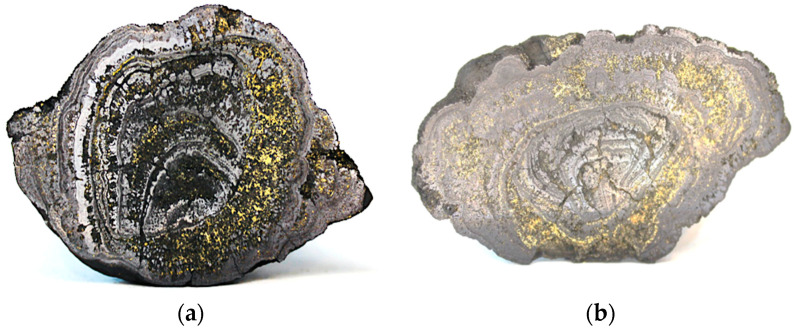
Example views of the chipping (**a**) and smooth (**b**) cross-sections of polymetallic concretions formed using a high-pressure AWJ.

**Figure 4 materials-15-08228-f004:**
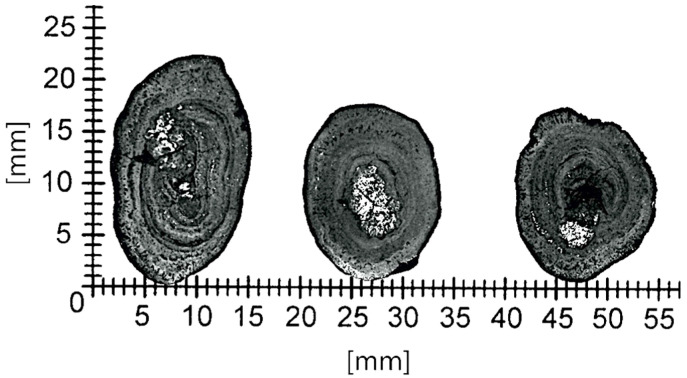
Example images of crystallographic specimens of polymetallic nodules.

**Figure 5 materials-15-08228-f005:**
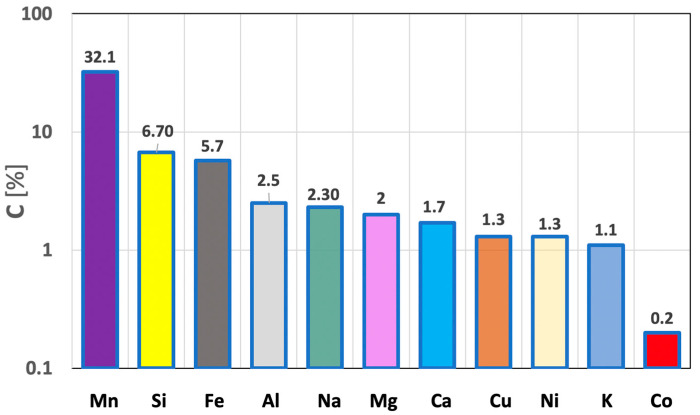
Chemical compound of polymetallic nodules from Clarion-Clipperton area.

**Figure 6 materials-15-08228-f006:**
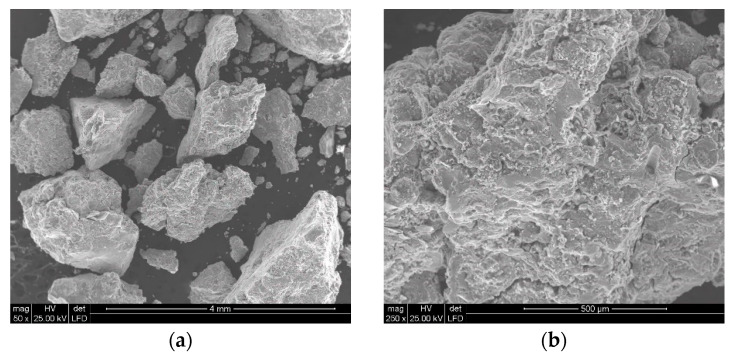
Example SEM images of polymetallic nodule particles (**a**), along with a view of their surface development (**b**), crushed in a jaw crusher.

**Figure 7 materials-15-08228-f007:**
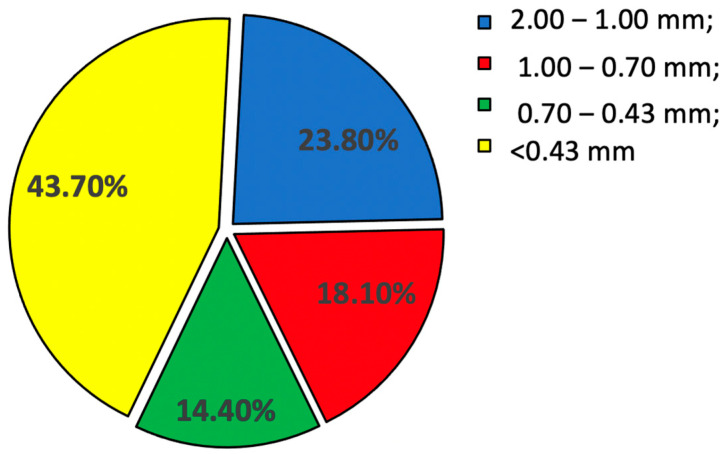
Percentage mass content of polymetallic nodule particles of different sizes, crushed in a jaw crusher.

**Figure 8 materials-15-08228-f008:**
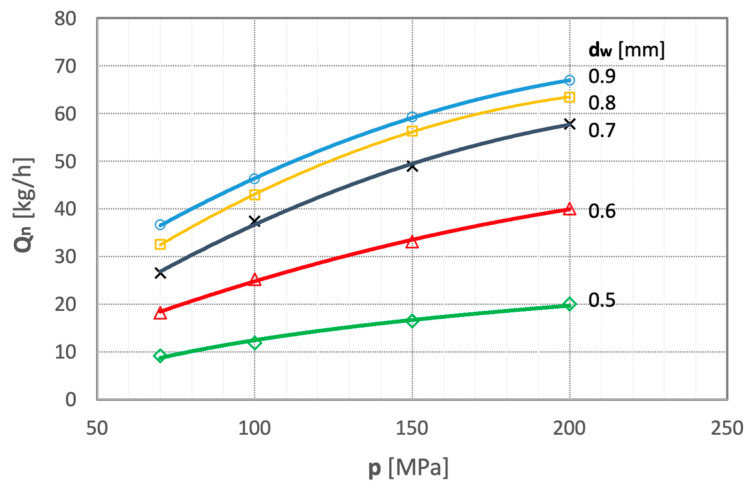
Plots of comminution efficiency of polymetallic nodules as a function of water pressure for several water jets of different diameters (for d_h_ = 2.4 mm and s = 10 mm).

**Figure 9 materials-15-08228-f009:**
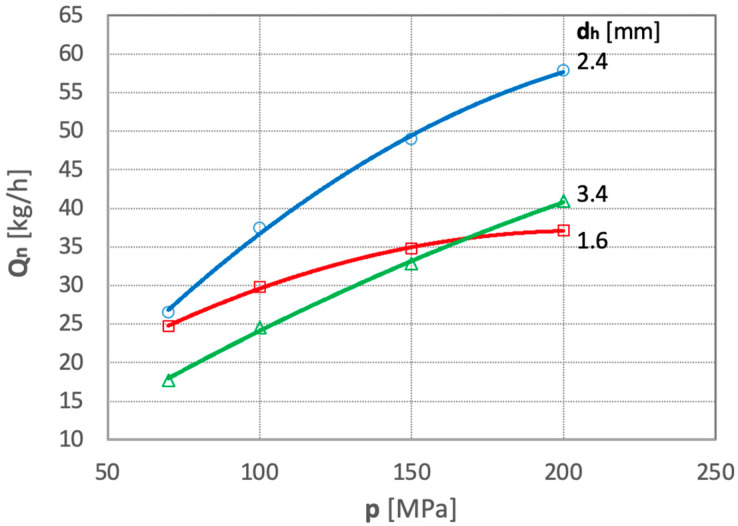
Dependence of the efficiency of polymetallic nodule comminution as a function of water pressure for several comminuting-homogenizing tubes of different diameters (for d_w_ = 0.7 mm and s = 10 mm).

**Figure 10 materials-15-08228-f010:**
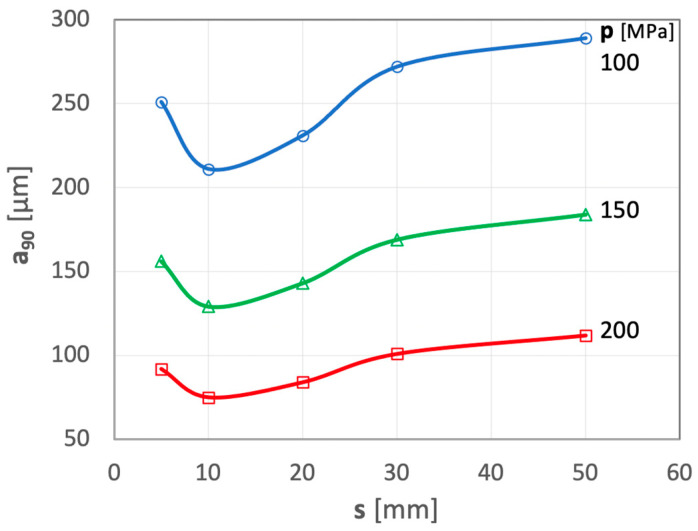
Influence of distance s from the comminuting-homogenizing tube outlet to the comminuting disk on the size of the average dimension a_90_ of the comminuted polymetallic nodule particles. Test conditions: d_w_ = 0.7 mm, d_h_ = 2.4 mm.

**Figure 11 materials-15-08228-f011:**
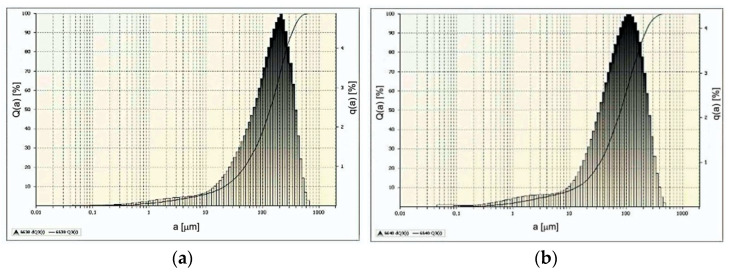
Example distributions of polymetallic nodule particles comminuted in a hydro-jet mill (for d_w_ = 0.7 mm, d_h_ = 2.4 mm, s = 10 mm) at water jet pressures: (**a**) 70 MPa, (**b**) 100 MPa.

**Figure 12 materials-15-08228-f012:**
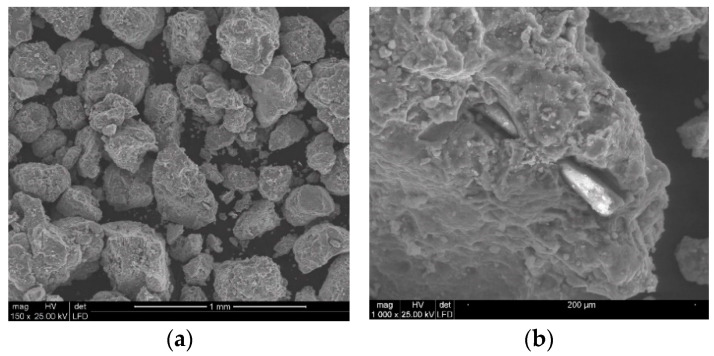
Example SEM images of concretion particles comminuted in a waterjet mill (for d_w_ = 0.7 mm, d_h_ = 2.4 mm, s = 10 mm, p = 70 MPa): (**a**) general view, (**b**) surface development.

**Figure 13 materials-15-08228-f013:**
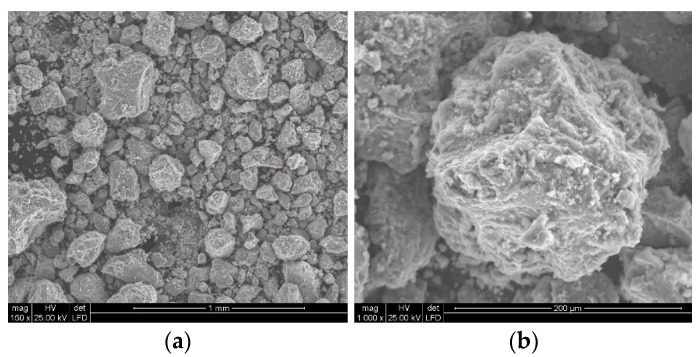
Example SEM images of particles of concretions comminuted in a waterjet mill (for d_w_ = 0.7 mm, d_h_ = 2.4 mm, s = 10 mm, p = 100 MPa): (**a**) general view, (**b**) surface development.

**Figure 14 materials-15-08228-f014:**
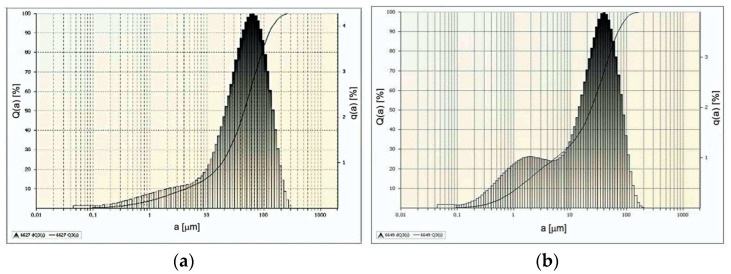
Example particle distributions of polymetallic nodules comminuted in a hydro-jet mill (for d_w_ = 0.7 mm, d_h_ = 2.4 mm, s = 10 mm) at water jet pressures: (**a**) 150 MPa, (**b**) 200 MPa.

**Figure 15 materials-15-08228-f015:**
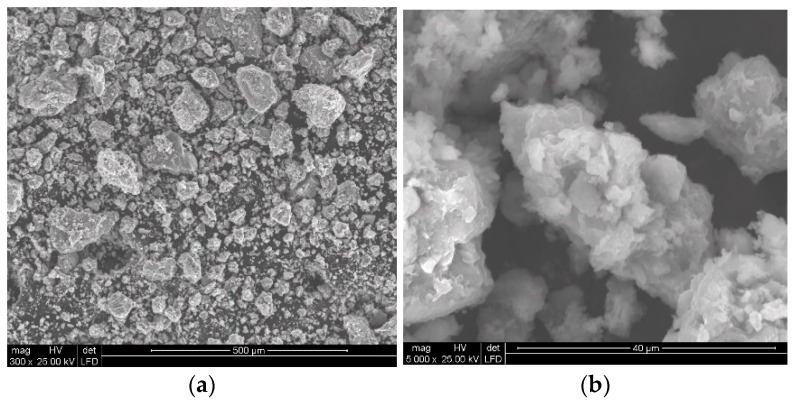
Example SEM images of the particles of concretions comminuted in a waterjet mill (for d_w_ = 0.7 mm, d_h_ = 2.4 mm, s = 10 mm, p = 150 MPa): (**a**) general view, (**b**) surface development.

**Figure 16 materials-15-08228-f016:**
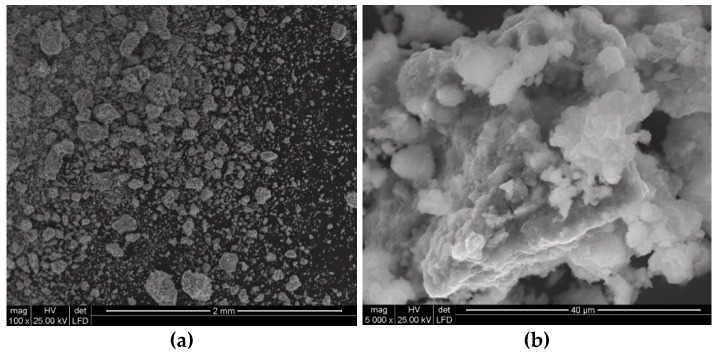
Example SEM images of concretion particles comminuted in a waterjet mill (for d_w_ = 0.7 mm, d_h_ = 2.4 mm, s = 10 mm, p = 200 MPa): (**a**) general view, (**b**) surface development.

**Figure 17 materials-15-08228-f017:**
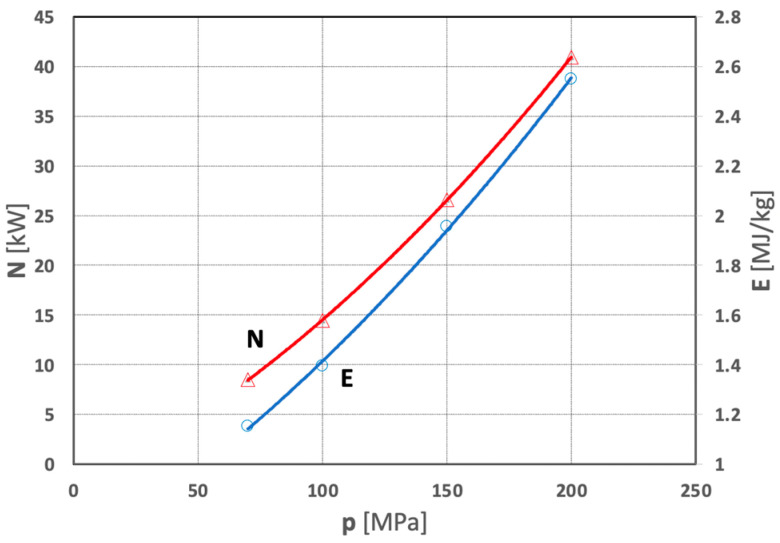
Influence of water jet pressure on water jet power (N) and unit energy expenditure (E) of water jet comminution of concretions (for d_w_ = 0.7 mm, d_h_ = 2.4 mm, s = 10 mm).

**Figure 18 materials-15-08228-f018:**
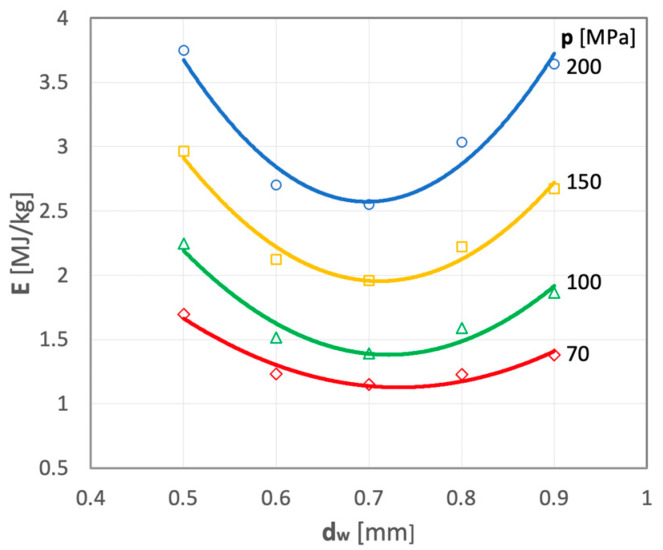
Specific energy expenditure (E) required to comminute polymetallic nodules as a function of water nozzle diameter (d_w_) for selected water pressure levels (for: d_h_ = 2.4 mm and s = 10 mm).

**Figure 19 materials-15-08228-f019:**
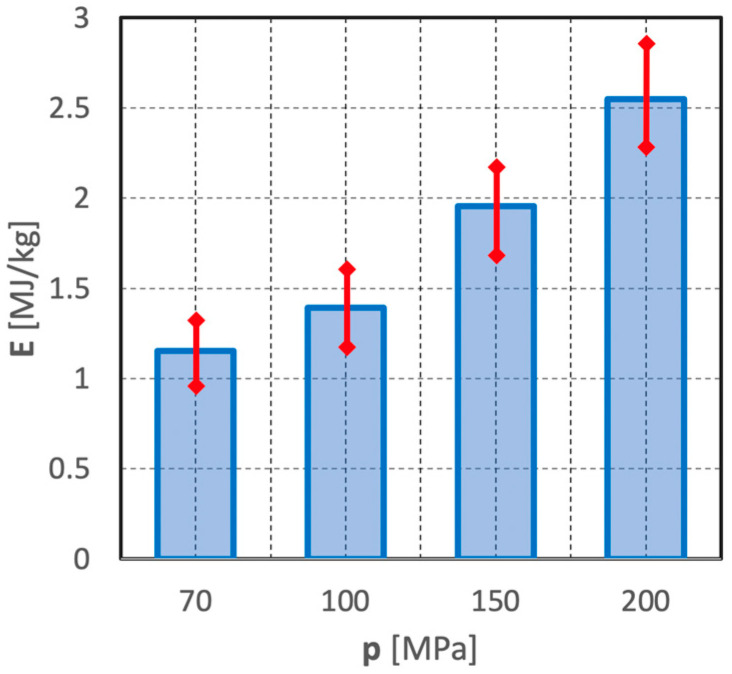
Effect of water jet pressure on the unit energy expenditure of polymetallic concretions comminution in a hydro-jet mill (for d_w_ = 0.7 mm, d_h_ = 2.4 mm, s = 10 mm).

**Figure 20 materials-15-08228-f020:**
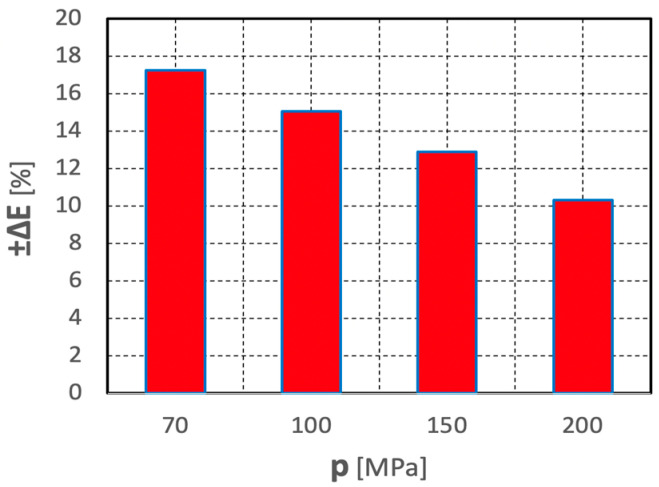
Effect of water jet pressure on the spread of relative values of unit energy inputs during the comminution of polymetallic concretions in a hydro-jet mill (for d_w_ = 0.7 mm, d_h_ = 2.4 mm, s = 10 mm).

**Figure 21 materials-15-08228-f021:**
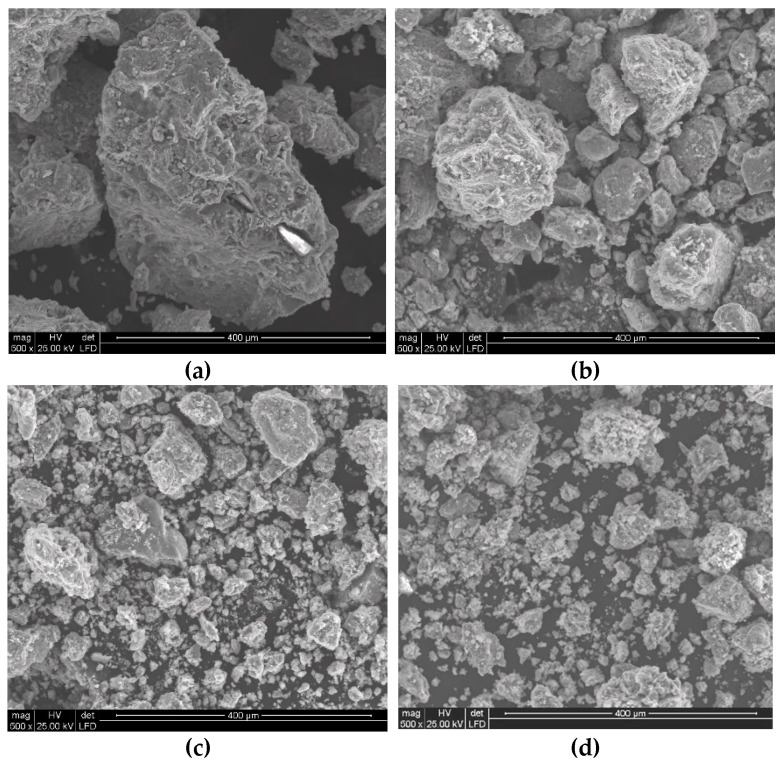
Example SEM images of concretion particles comminuted in a hydro-jet mill (for d_w_ = 0.7 mm, d_h_ = 2.4 mm, s = 10 mm) at water jet pressures: (**a**) 70 MPa, (**b**) 100 MPa, (**c**) 150 MPa, (**d**) 200 MPa, and at equal magnification (500×).

**Figure 22 materials-15-08228-f022:**
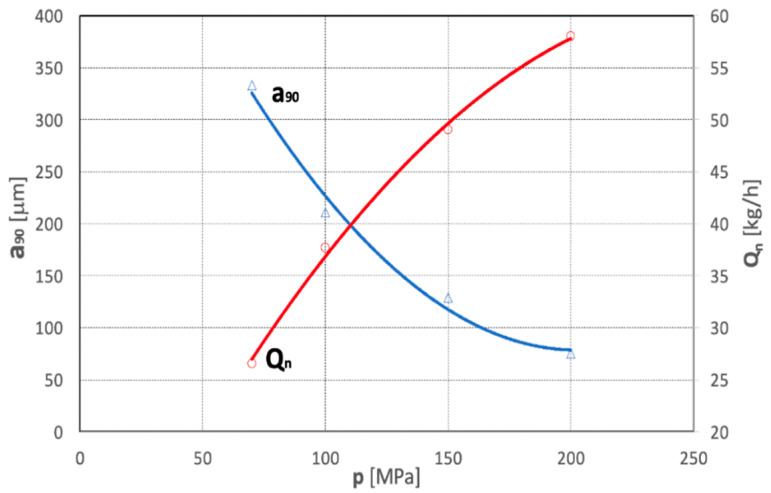
Influence of water jet pressure on the particle size (a_90_) of comminuted concretions and efficiency (Q_n_) of their water jet comminution process (for d_w_ = 0.7 mm, d_h_ = 2.4 mm, s = 10 mm).

**Figure 23 materials-15-08228-f023:**
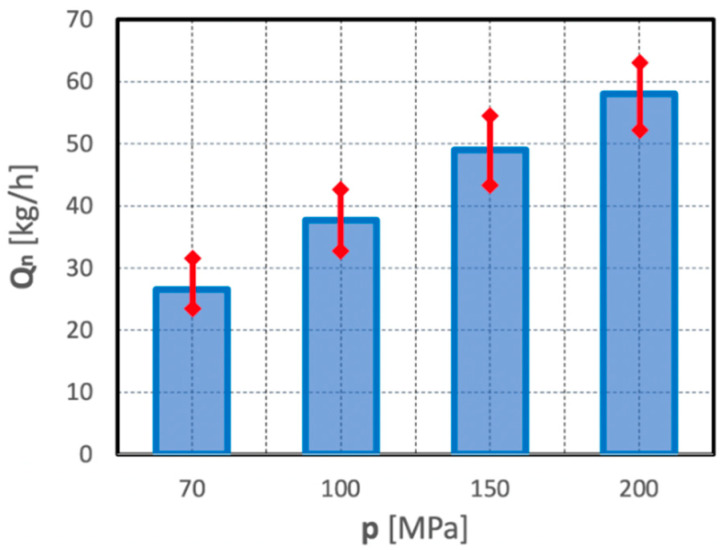
Effect of water jet pressure on the efficiency of the comminution process of polymetallic concretions in a hydro-jet mill (for d_w_ = 0.7 mm, d_h_ = 2.4 mm, s = 10 mm).

**Figure 24 materials-15-08228-f024:**
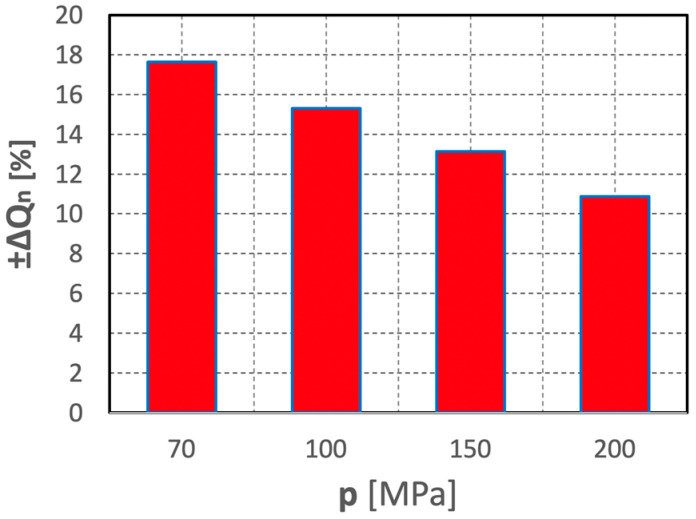
Influence of water jet pressure on the distributions of relative concretion comminution values, during the comminution in a hydro-jet mill (for d_w_ = 0.7 mm, d_h_ = 2.4 mm, s = 10 mm).

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
