# Peer review of "Comminution of Polymetallic Nodules with a High-Pressure Water Jet"

_materials, 2022, doi:10.3390/ma15228228_

Round 1

Reviewer 1 Report

 This paper mainly analyzed the comminution effect of polymetallic nodules under various high pressure water jet. In my opinion, this paper is not a complete scientific research paper. In this paper, only the morphology and the dimension of the polymetallic nodule particles under different water jet pressure is compared, where some of the results are obvious. This paper lacks reasonable processing parameter optimization analysis or in-depth scientific result analysis.

Other comments are listed as follows.

(1) This paper is not a review paper. There are too many references, some irrelevant ones can be deleted.

(2) The content of Introduction is too lengthy. In addition, it should introduce  development of different comminution process or their advantage and disadvantage in detail.

(3) The proper optimization method, such as the response surface method, should be adopted to determine the optimized process parameters, including the water nozzle diameter, homogenizing nozzle diameter and the distance.

(4) More results, including the influence of each processing parameter on the communution effect and the performance changes, should be analyzed in detail.

(5) The format of the references is in consistent.

(6) The language should be improved to increase readability.

In general, this paper is not well organized, it should be rejected at this stage.

Author Response

Dear Reviewer,

Thank you for your time contributed to improve our manuscript entitled: Comminution of polymetallic nodules with a high-pressure water jet. We have carefully analyzed your critical comments and below I include our answers to respective issues. Before that however, I am going to point out my background and specific perspective of the presented information. I am a full professor of mechanical engineering that deals with HPWJ for over two decades. But last one I have spent in the copper mining industry as well in geological institute leading the group aiming at new technological approach to resolve important issues e.g. of energetic minerals, among them foreseeing the future role of deep-sea mining.

What concerns to your comments given below:

(1) This paper is not a review paper. There are too many references, some irrelevant ones can be deleted

and (2) The content of “Introduction” is too lengthy. In addition, it should introduce  development of different comminution process or their advantage and disadvantage in detail.

Your comment might be wright in the sense of the clarity of the information given to the readers of most issues connected to the problem of DSM in a proportional way. However, having in mind some experience in that matter, I recall some other papers and respective suggestions e.g. from the MDPI editor to fully introduce up to date references. Having this in mind as well as knowing the interest in DSM, I just tried to summarize the whole scope of activity in given manuscript. The given introduction might be considered as too extensive one in this sense; therefore, I have made an appropriate reassumption limiting the number and the description of cited papers to the most relevant by giving now a half of previously presented ones.

Additionally, I have pointed out some aspect of different comminution processes as well by presenting other works [45-50] given in Chapter 2. Equipment, apparatus and test conditions, but detailed information concerning advantages and disadvantages in details might have open another chapter. It was characterized in other grant report of mine: Borkowski P.: Determination the feasibility of intensifying high-pressure water jet as an alternative tool for comminuting copper ore in KGHM Polska Miedź S.A. Grant no. B/07/5030.001 from KGHM Polish Copper S.A. Wrocław. 2015 (in Polish).

(3) The proper optimization method, such as the response surface method, should be adopted to determine the optimized process parameters, including the water nozzle diameter, homogenizing nozzle diameter and the distance.

and (4) More results, including the influence of each processing parameter on the communution effect and the performance changes, should be analyzed in detail.

Answering to above questions (3) and (4), the optimization of the process was published in my previous works, therefore intentionally I have described optimum setup in the first version of the manuscript. After reading your remarks, I have added additional chapter to enclosed revised manuscript (5. Determination of optimum parameters for comminution of polymetallic nodules by water jet) that deal with the problem of chosen technological parameters optimization by detailed information and respective illustrations.

(5) The format of the references is in consistent.

If I properly understood your remark, by changing the introduction section, the format of references also has changed and is consistent. All the references meet the given standard of editor (as given in template info and doublechecked in current Materials Journal, vol. 15).

(6) The language should be improved to increase readability.

What considers English language used in preparation of the manuscript, it was given a help in translation from a professional staff, I used to cooperate with for a while and his previous few works (translations) were positively judged by publishing e.g. in MDPI Minerals in recent years.

In general, this paper is not well organized, it should be rejected at this stage.

In general, I hope you will find enclosed corrected manuscript improved in regards to given issues that will increase the interest of potential readers. 

Thank you once again for your review.

Sincerely,

Przemyslaw Borkowski

Reviewer 2 Report

The manuscript is related to the interesting topic that is the deep mining from the seabed. However, the intrinsic topic of the manuscript is the comminution of material particles by water jet. Therefore, the amount of the cited articles aimed at problems of sea mining is disproportionately large. Many cited articles deal with the similar or identical topics and, in spite of the fact that they document the extent of the problems and the urgency of their solution; they are redundant for the requirements of the introduction to the presented research. On the other hand, the intrinsic problem studied by authors, the comminution of material by water jet, has only three citations of the own publications. The comminution of material particles by pure water jets has been studied by other authors several times and, therefore, some of those publications should be cited. It is likely that some of them even served as an inspiration for the design of your own mill. The other comments and suggestions are presented hereafter:

1) It is not a good practice in research articles to cite more than two sources without an analysis of how each of them contributed significantly to solving the problem. Moreover, the reduction of the number of the articles aimed at the same problems is recommended. There are cited at least 17 articles from the International Seabed Authority (Kingston, Jamaica) in 2001, then at least 15 articles from the International Seabed Authority's Workshop (Kingston, Jamaica) in 2006 and at least 13 articles from the International Seabed Authority (Kingston, Jamaica) in 2017. Many of these articles deal with the same problems as the ones cited from journals or some proceedings and books. As the problems described in these documents are just illustrative to the presented research aim, it can be selected just one citation to each partial problematics mentioned in the Introduction.

2) In order to maintain the sequence of references, it is necessary to move reference [21] to position [19] (better said to modify the reduced bibliography accordingly so that the reference to this article does not precede those listed in the section References with the lower sequence numbers).

3) Line 61: The REO metal identification is unclear.

4) Some of the information presented in the Introduction is interesting (e.g. in lines 73-81), but without any connection to the problem presented as the topic of this particular manuscript.

5) The term homogenization (or comminution-homogenization) "nozzle" is improper. From the physical point of view, only the element converting the static energy (pressure) into the kinetic energy of the flowing liquid can be marked as nozzle. The element in the mill is rather the focusing tube (or its very similar analogy). Therefore, the marking "nozzle" is necessary to change into some other word in this context (homogenizing or comminuting element).

6) Figure 1(b) shows the internal arrangement of the mill. However, no details are presented regarding the "comminuting disk". It is necessary to present, if it is static or rotating disc (and with which frequency), what is the material of the disc surface, how much it pollutes the product etc.

7) Figure 2 and Figure 4: The units are indispensable.

8) The separation of the value and unit (or parts of the equation) by the line break is present many times throughout the manuscript. Some examples: lines 267-268, 281-282, 301-302, 303-304, 330-331, 339-340, 381-382, 406-407, 443-444, 492-493, 496-497, 503-504. It is necessary to prevent such cases using inseparable space or fixed hyphen in the relevant links.

9) The results presented in Figures 8 and 11 are to be compared with results of other researchers. Two examples can be the article at the address http://transactions.fs.vsb.cz/2007-1/1534.pdf or the one at this address https://doi.org/10.1016/j.minpro.2010.03.003. The respective comparison can confirm that the most important disintegration takes place in the mixing chamber and the focussing tube and only a small amount of disintegration can be attributed to the impact on the disc.

10) The equation in the line 489 (in Conclusions) does not correspond with the Equation (3) in line 358. What is the source of this discrepancy?

11) Conclusions presented in lines 490-493 and 494-497 have no justification in the presented data and analyses. Moreover, the number 2 at the pressure p needs to be in the superscript format.

12) The statement in the lines 508-509 cannot be presented as a conclusion (at least not in the presented formulation).

13) The proofreading by a native speaker is strongly recommended.

14) The format of some cited articles in the part References is not in the correct format. Namely presentation of authors names as in the lines 650, 653, 678, 700, 703, 750. The abbreviation "Surname et al." is usable for better resolution of the respective link in the text, not in the list of referred works.

Author Response

Dear Reviewer,

Thank you for your time contributed to improve our manuscript entitled: Comminution of polymetallic nodules with a high-pressure water jet. We have carefully analyzed your comments and below you will find our discussion to respective issues. Before that however, I am going to point out my background and specific perspective of the presented information. I am a full professor of mechanical engineering that deals with HPWJ for over two decades. But last one I have spent in the copper mining industry as well in geological institute leading the group aiming at new technological approach to resolve important issues e.g. of energetic minerals, among them foreseeing the future role of deep-sea mining.

What concerns to your comments given below:

1) It is not a good practice in research articles to cite more than two sources without an analysis of how each of them contributed significantly to solving the problem. Moreover, the reduction of the number of the articles aimed at the same problems is recommended. There are cited at least 17 articles from the International Seabed Authority (Kingston, Jamaica) in 2001, then at least 15 articles from the International Seabed Authority's Workshop (Kingston, Jamaica) in 2006 and at least 13 articles from the International Seabed Authority (Kingston, Jamaica) in 2017. Many of these articles deal with the same problems as the ones cited from journals or some proceedings and books. As the problems described in these documents are just illustrative to the presented research aim, it can be selected just one citation to each partial problematics mentioned in the Introduction.

Your comment might be wright in the sense of the general clarity of information typically directed to potential readers. The good practice you mentioned seems to be more general. If you analyze for example current MDPI Materials, Volume 15, Issue 1 (January-1 2022) – 396 articles, three first papers given there, or even cover story paper, one can find different practice including cited 11 different papers in one sentence, without additional discussion. Nether the less, being a conciliatory person, I have changed a lot in this matter directly in the introduction section and consequently in the next chapters. You will also find now suggested two-source practice prepared in order to increase the quality of the paper.

All kind aspects of all activity can be discussed depending on the reviewer’s point of view. That is a common practice. However, having in mind some experience in that matter, I recall some other papers of mine and suggestions e.g. from the MDPI editor to fully introduce the readers with up to date references. Having this in mind as well as knowing the interest in DSM, I just tried to summarize the whole scope of activity in first version of the manuscript. The given introduction might be considered as too extensive one in this sense; therefore, I have made an appropriate reassumption limiting the number and the description of cited papers to the most relevant by giving now a half of previously presented ones.

2) In order to maintain the sequence of references, it is necessary to move reference [21] to position [19] (better said to modify the reduced bibliography accordingly so that the reference to this article does not precede those listed in the section References with the lower sequence numbers).

As I mentioned in above discussion, references list was modified according to changes made in the text of revied manuscript.

3) Line 61: The REO metal identification is unclear.

The REO stand for rare earth oxidants and was added to nomenclature section at the end of the manuscript.

4) Some of the information presented in the Introduction is interesting (e.g. in lines 73-81), but without any connection to the problem presented as the topic of this particular manuscript.

As mentioned previously, a significant correction to the introduction section was made in order to clarify and simplify the information.

5) The term homogenization (or comminution-homogenization) "nozzle" is improper. From the physical point of view, only the element converting the static energy (pressure) into the kinetic energy of the flowing liquid can be marked as nozzle. The element in the mill is rather the focusing tube (or its very similar analogy). Therefore, the marking "nozzle" is necessary to change into some other word in this context (homogenizing or comminuting element).

Thank you for pointing out obvious mistake. You are wright in this aspect. Usually, one may call it a nozzle but it is unproper in the terms of the function of the mill construction. That is why we have changed the name into comminuting-homogenizing tube consequently in the whole text.

6) Figure 1(b) shows the internal arrangement of the mill. However, no details are presented regarding the "comminuting disk". It is necessary to present, if it is static or rotating disc (and with which frequency), what is the material of the disc surface, how much it pollutes the product etc.

Concerning remark no. 6, we have added relevant information to the section 2. Equipment, apparatus and test conditions (lines 94-100) by describing internals characteristics of the mill. “A schematic diagram of the construction of such a mill is shown in Fig. 1. Its grinding elements comprise a comminuting-homogenizing tube (4) and a comminuting disk (5), which are made of sintered tungsten carbide. The opening of the comminuting-homogenizing tube has an inlet part with a threshold edge, where the feed is initially comminuted. The final comminution of the ore is the result of collisions with the surface of the comminuting disk, which is static but can be rotated by an appropriate angle as its surface erodes”.

7) Figure 2 and Figure 4: The units are indispensable.

Thank you for pointing out obviously omitted information. It was corrected to enclosed revied version of the manuscript.

8) The separation of the value and unit (or parts of the equation) by the line break is present many times throughout the manuscript. Some examples: lines 267-268, 281-282, 301-302, 303-304, 330-331, 339-340, 381-382, 406-407, 443-444, 492-493, 496-497, 503-504. It is necessary to prevent such cases using inseparable space or fixed hyphen in the relevant links.

I have made correction throughout the text. Thank you.

9) The results presented in Figures 8 and 11 are to be compared with results of other researchers. Two examples can be the article at the address http://transactions.fs.vsb.cz/2007-1/1534.pdf or the one at this address https://doi.org/10.1016/j.minpro.2010.03.003. The respective comparison can confirm that the most important disintegration takes place in the mixing chamber and the focussing tube and only a small amount of disintegration can be attributed to the impact on the disc.

Additionally, I have pointed out some aspect of different comminution processes as well by presenting other works [45-50] given in Chapter 2. Equipment, apparatus and test conditions but detailed information concerning advantages and disadvantages in details might have open another chapter and consequently was characterized in other e.g. grant report of mine: Borkowski P.: Determination the feasibility of intensifying high-pressure water jet as an alternative tool for comminuting copper ore in KGHM Polska Miedź S.A. Grant no. B/07/5030.001 from KGHM Polish Cupper S.A. Wrocław. 2015 (in Polish) or in M. Bielecki: Study of the effect of the parameters of a high-pressure waterjet mill on the comminution efficiency of materials. Doctoral Dissertation. Koszalin University of Technology, 2013 (in Polish).

Presented in your remark two different papers included in the reference section, considers different constructions of a head/mill and obviously leads to given conclusions.

10) The equation in the line 489 (in Conclusions) does not correspond with the Equation (3) in line What is the source of this discrepancy?

Thank you. That was a technical mistake. Now all equations correspond with respective information in the Conclusion section.

11) Conclusions presented in lines 490-493 and 494-497 have no justification in the presented data and analyses. Moreover, the number 2 at the pressure p needs to be in the superscript format.

There was indeed no representation of the conclusion in the text. My mistake. I have described the issue now by additional verses (and Fig. 22.) starting from 437 – 461 (page 15).

Moreover, I have added additional chapter to enclosed revised manuscript (5. Determination of optimum parameters for comminution of polymetallic nodules by water jet) that deal with the problem of chosen technological parameters optimization by detailed information and respective illustrations.

12) The statement in the lines 508-509 cannot be presented as a conclusion (at least not in the presented formulation).

By giving additional explanation to the enclosed corrected text of our manuscript, including new chapter that determines the optimum parameters for comminution of nodules by water jet, and other information taken as a whole text being a consequence of experimental approach, the given conclusion saying at the end that “For the above reasons, the water jet method of comminution of such concretions is very promising” is wright from the authors perspective. We believe the given technique can be implemented in the processing part of DSM (initial insitu comminution that resolves problematic vertical transportation issues from the bottom to the surface). Moreover, such a method, without any doubt is environmentally friendly to the sensitive ecosystem and relatively technically easy to maintain comparing to mechanical operations that can only be operated remotely from the surface. In such a way, taking additionally authors experience in the field (prof. T. Abramowski, additionally is a general director of The Interoceanmetal Joint Organization (IOM) we stand for correctness of given conclusion.

13) The proofreading by a native speaker is strongly recommended.

What considers English language used in preparation of the manuscript, it was given a help in translation from a professional staff, I used to cooperate with for a while and his previous few works (translations) were positively judged by publishing e.g. in MDPI Minerals in recent years.

14) The format of some cited articles in the part References is not in the correct format. Namely presentation of authors names as in the lines 650, 653, 678, 700, 703, 750. The abbreviation "Surname et al." is usable for better resolution of the respective link in the text, not in the list of referred works.

All the references were checked for eventual name abbreviations and corrected throughout the chapter.

Thank you once again for your review that surely improved our manuscript.

Sincerely,

Przemyslaw Borkowski

Round 2

Reviewer 1 Report

The authors have carefully revised the manuscript according to my comments, so at this stage, it can be accepted.

Author Response

Dear Reviewer,

Once again, I would like to thank you for your efforts to improve our manuscript. In addition, it should be mentioned that it was finally checked by a native speaker for correctness. 

Sincerely,

Przemyslaw Borkowski

Reviewer 2 Report

Most of the shortcomings have been eliminated, but a few problem areas remain:

1) The improper term "nozzle" instead of "tube" remained in the lines 16, 17 (Abstract), 515 and 516 (Conclusions) and in lines 568 and 572 (Nomenclature).

2) The improper separation of "dw=" and "0.7 mm" by the line break can be find in the end of lines 15, 291, 293, 324, 410, 488. The inseparable space implemented instead of the normal space can be used in Word program holding down the shift and ctrl keys and pressing the space bar.

3)The format of equations should be unified, i.e. the left margin of the equation should be on the identical position as well as the right margin of the identifying number. Moreover, there are some senseless dots in line 444.

4) Line 604: It seems to me that name "COMRA" is unsuitably all in the capital letters.

5) It is recommended to unify the format of the "DOI" links in References to the format https://doi.org/number of publication

Author Response

Dear Reviewer,

Once again, I would like to thank you for your efforts to improve our manuscript. Regarding the last 5 technical points, we have made corrections to the text.  In addition, it should be mentioned that it was finally checked by a native speaker for correctness.   

Sincerely,

Przemyslaw Borkowski